# The Prevalence of and Trend in Drug Use among Adolescents in Mississippi and the United States: Youth Risk Behavior Surveillance System (YRBSS) 2001–2021

**DOI:** 10.3390/ijerph21070919

**Published:** 2024-07-14

**Authors:** Zhen Zhang, Amal K. Mitra, Julie A. Schroeder, Lei Zhang

**Affiliations:** 1Department of Epidemiology and Biostatistics, School of Public Health, College of Health Sciences, Jackson State University, Jackson, MS 39213, USA; zhen.zhang@jsums.edu; 2Department of Public Health, Julia Jones Matthews School of Population and Public Health, Texas Tech University Health Sciences Center, Abilene, TX 79601, USA; 3School of Social Work, College of Health Sciences, Jackson State University, Jackson, MS 39213, USA; julie.a.schroeder@jsums.edu; 4School of Nursing, University of Mississippi Medical Center, Jackson, MS 39216, USA; lzhang2@umc.edu

**Keywords:** prevalence, trend, marijuana, inhalants, heroin, methamphetamines, illegal drug, YRBSS

## Abstract

Mississippi youth are demographically unique compared to those of the nation. The aim of the study was to examine the drug use among adolescents in Mississippi compared to that in the US, which included determining prevalence and trends in drug use as well as drugs on school property and estimating the differences in drug use prevalence by gender and by race. National and Mississippi Youth Risk Behavior Surveillance System (YRBSS) data from 2001 to 2021 were obtained for analysis. Summary statistics, prevalence ratio, and survey Chi-squared tests of independence statistics were generated for the comparison for all students, and by gender and race separately. Trend analysis was conducted using logistic regression combined with joinpoint regression. The six survey questions being studied were the following: have you ever used marijuana, an inhalant, heroin, methamphetamines, or injected drugs, and were you offered, sold, or given an illegal drug on school property during the last 12 months. Survey packages in R were used to account for the complex sampling design of YRBSS data. On the national level, all six drug-related risk behaviors being studied showed a significant decrease from 2001 to 2021. In Mississippi, however, only “ever used marijuana” showed a decrease trend, while three remain unchanged, and two increased. The 2021 YRBSS data show that Mississippi adolescents exhibited a significantly higher prevalence of drug use, and are more likely to be offered, sold or given an illegal drug on school property. This research showed detailed findings on drug use-related issues in Mississippi, which is alarming. This poses an important challenge for public health in Mississippi and sounds an urgent call for drug use intervention among Mississippi adolescents. More concerted actions at the community, school and government level are needed for reducing youth drug use and controlling the drug traffic on school property.

## 1. Introduction

In the United States, approximately 15% of high school students reported having any illicit drug use, including marijuana, heroin, methamphetamines, inhalants, hallucinogens or ecstasy [1]. According to the National Institute on Drug Abuse (NIDA), adolescents’ rate of using illicit drugs in 2023 showed a downward trend compared to the pre-pandemic period, probably attributable to school closing and social distancing during the COVID-19 pandemic [2]. Youth drug use is directly linked with increased sexual risk behaviors, experience of violence, adverse mental health and suicidality among adolescents and youths [3]. Data suggests that young students, as they transition to colleges, are influenced by their peers to undertake risky behaviors including drug use, suicidal thoughts, and non-suicidal self-injury [4]. Studies further emphasized the risk of substance-induced psychosis, leading to “clinically significant impairment” and suicidality [5,6].

Prevalence and the patterns of adolescents’ drug use are often correlated with concurrent other high-risk behaviors, such as engaging in unprotected sex, sex with multiple partners, and having intimate partners who use drugs [7,8]. In 2023, the prevalence of illicit drug use in the past year increased with increasing age of the youths—10.9% in eighth graders, 19.8% in 10th graders, and 31.2% in 12th graders [8]. According to the National Survey on Drug Use and Health data (2004–2019), the pattern of drug use initiation varies by race and ethnicity [9]. In an earlier study of trends and patterns relating to youth substance uses, lifetime use of marijuana decreased in the U.S. during 2013–2019 [10]. In 2019, 21.7% reported current marijuana use, while 13.7% reported current binge drinking, and 7.2% reported current prescription opioid misuse.

Youth drug use is also associated with sexual risk behaviors that make young people vulnerable to sexually transmitted infections (STIs), increased prevalence of teenage pregnancy, and poor mental health [3,8]. Unfortunately, Mississippi has the highest combined rates of major STIs [11] and is among the states with high teen pregnancy rates [12]. 

There are several gaps in the knowledge about youth drug use: (1) information about the recent prevalence and trend regarding different drugs is lacking in this age group; (2) Mississippi having issues of poor health outcomes, it was imperative to conduct an in-depth study to understand if these adverse health outcomes are some way or the other related to youth drug-use risky behaviors; and (3) there is also gap in the knowledge in the prevalence ratio between the rates of different drugs commonly used by the youth and adolescents in Mississippi and the nation. Therefore, the objectives of this study were to examine the drug use among adolescents in Mississippi compared to that in the US, which included comparison of prevalence and trends in drug use as well as drugs on school property.

## 2. Materials and Methods

### 2.1. Data Source

National and Mississippi YRBSS data were used for this study. YRBSS is a set of surveys that track the major health risk behavior factors among adolescents. YRBSS has two components: the first component is the national survey conducted by CDC that includes high school students from both private and public schools of 50 US states and the District of Columbia. The second component is surveys conducted by the Department of Health and Education at the state, tribal, territorial or local levels among the respective public high school students. Data used for this study were drawn from both sources. YRBSS is a three-stage complex sample design to ensure representative samples are being collected. The first stage determines the primary sampling unit (PSU) to be at the county (or comparable geographic unit) level. The PSUs are further categorized into strata according to their metropolitan location and racial composition. For the second stage, the secondary sampling unit (SSU) is defined at the school level. Both PSUs and SSUs are sampled with probability proportional to overall school enrollment size. The third stage is random sampling of one or two classrooms in each of grades 9–12 of the selected schools. YRBSS is conducted biannually [13].

### 2.2. Measurements

Student’s responses to drug use-related YRBSS survey questions, which are expressed as ordinal or nominal data, are re-coded as binary data (Yes, No) for analysis. Following, are the six data points used for this study: QN45—ever used marijuana; QN51—ever used inhalants; QN52—ever used heroin; QN53—ever used methamphetamines; QN55—ever injected any illegal drug; QN56—were offered, sold, or given an illegal drug on school property.

Survey questions relevant to this study also included: gender (male, female) and race/ethnicity (Non-Hispanic American Indian or Alaska Native, Non-Hispanic Asian, Non-Hispanic Black, Hispanic, Non-Hispanic Native Hawaiian or Other Pacific Islander, Non-Hispanic White and Non-Hispanic Multiple race). The race variable used in this study has four values (White, Black, Hispanic and Multiple-race non-Hispanic) [14,15].

### 2.3. Statistical Analysis

Drug use-prevalence estimates and their 95% confidence intervals (*CI*s) were calculated and compared between national and Mississippi adolescents for the survey year 2021. Comparison was carried out as a whole, as well as by each gender subgroup and each race subgroup. We used the survey package in R to produce prevalence ratios and *p* values. In the rare instances where the 95% *CI*s overlap but the *p*-value is less than 0.05, we reported the difference as statistically significant [16,17,18].

For the trend analysis, we took the approach as prescribed by CDC. First, we used logistic regression models to detect linear and/or non-linear (quadratic or cubic) trends. Log odds of each variable were estimated as a function of time, time in quadratic term, and time in cubic term, respectively. Time variables were created by coding each year with orthogonal coefficients, and they were treated as continuous covariate in the models. Only the highest-order time variable in the model was considered valid. Secondly, if only a linear-year contrast term was found to be significant, then the associated beta and *p*-value for that term was used to determine the direction and significance of the trend [19]. If quadratic and/or cubic changes were detected, we took the next step to calculate the adjusted (by sex, race and grade) prevalence and standard error by survey year and export these values into Joinpoint software (version 5.1.0) (Available online: https://surveillance.cancer.gov/joinpoint/, accessed on 2 June 2024). to determine the critical values of the joints for trend segments. The permutation test was used to determine the number of joinpoints and the parametric method was applied to calculate the confidence intervals for Annual Percentage Change (APC) of the resulting segments and Average Annual Percentage Change (AAPC) of the whole trend [20,21]. 

IBM SPSS Modeler (version 18.4) (IBM, New York, NY, USA) was used to produce sample characteristic statistics. Statistical software R (version 4.4.0) (Available online: https://www.r-project.org/, accessed on 2 June 2024) was used for summary statistics and logistic regression models. R with its survey packages was identified by CDC as an appropriate software that is capable of accounting for the complex sampling design of YRBSS data [2,13]. Joinpoint software (version 5.1.0) was downloaded with permission from National Cancer Institute and used for joinpoint regression analysis.

## 3. Results

### 3.1. Sample Characteristics

The 2021 YRBS sample sizes for US and Mississippi were 17,232 and 1747, respectively. Gender distribution for US and Mississippi indicates that female and male students were approximately equally represented: 47.3% and 49.2% for female in the US and Mississippi, respectively. Significant variance was observed in the racial distribution of the sample. The national YRBS sample consisted of 53.1% White, 13.5% Black, 18.8% Hispanic and 5.8% non-Hispanic multiple races, while the Mississippi YRBS sample consisted of 33.8% White, 47% Black, 8.9% Hispanic and 3.3% non-Hispanic multiple races. This difference reflects the underline racial demographics in Mississippi and the United States as a whole. Compared to the national sample, the Mississippi sample included more 9th graders (36.1% vs. 27.0%) and fewer 12th graders (17.5% vs. 22.3%), while the percentages for 10th and 11th graders were similar.

### 3.2. Comparison of Drug Use-Related Risk Behaviors between Mississippi and the US 

Table 1 shows the comparison of the prevalence of drug use-related risk behaviors between Mississippi and the United States in 2021. To the question “have you ever used marijuana?”, there was no significant difference between Mississippi (MS) and the United States (US) However, compared the US, MS adolescents had significantly higher prevalence for ever having used inhalants, heroin, methamphetamines, and drug injection and were more likely to be offered, sold, or given an illegal drug on school property. The prevalence ratio for the above variables was 0.9 (*p* = 0.19), 1.5 (*p* < 0.001), 3.3 (*p* < 0.001), 2.6 (*p* < 0.001), 2.7 (*p* < 0.001), and 1.5 (*p* < 0.001), respectively. Overall, Mississippi adolescents exhibited significantly greater risk behaviors related to drug use.

Further analysis of subgroups shows that, compared to the US, the elevated risk behaviors related to inhalants, heroin, methamphetamines, and illegal drug injection, and high school students who were offered, sold or given an illegal drug on school property in Mississippi were not all distributed evenly among female and male students, nor were they evenly distributed among the race categories. 

#### 3.2.1. Marijuana and Inhalant Use

As shown in Table 2, compared to the US, Mississippi youth marijuana use had no significant difference, either as a whole (*p* = 0.19), or by gender (*p* = 0.1 for female, *p* = 0.61 for male) or race (*p* = 0.31 for White, *p* = 0.06 for Black, *p* = 0.20 for Hispanic and *p* = 0.27 for Multiple race). Compared to the US, inhalant use was significantly greater for male students in Mississippi (12.8% vs. 6.8%, *p* <0.001), but not for female students (10.6% vs. 9.4%, *p* = 0.47). Compared to the US, inhalant use was significantly greater only for Black students in Mississippi (11.8% vs. 7.0%, *p* = 0.02).

#### 3.2.2. Heroin and Methamphetamine Use

As shown in Table 3, compared to the US, heroin use prevalence was significantly higher for both female (2.3% vs. 0.8%, *p* = 0.02) and male (5.9% vs. 1.6%, *p* <0.001) students in Mississippi. Compared to the US, heroin use prevalence in Mississippi was significantly higher for Black students (5.8% vs. 1.7%, *p* < 0.001) and Hispanic students (8.9% vs. 1.6%, *p* = 0.03), but significantly lower in Multiple-race students (0% vs. 1.5%, *p* <0.001) and showed no difference for White students (1.5% vs. 1.0%, *p* = 0.61). Compared to the US, risk behavior related to methamphetamine use in Mississippi was significantly more prevalent in male students (5.6% vs. 1.9%, *p* < 0.001), but not in female students (2.7% vs. 1.4%, *p* = 0.08). Compared to the US, methamphetamine use prevalence in Mississippi was significantly higher for Black students (5.4% vs. 2.0%, *p* = 0.01) and Hispanic students (10.7% vs. 2.3%, *p* = 0.01), but no significant difference was observed in White students (2.3% vs. 1.4% *p* = 0.33) or in Multiple-race students (2.0% vs. 2.2%, *p* = 0.92). 

#### 3.2.3. Illegal Drug Injection and Drug on School Property

As shown in Table 4, compared to the US, risk behavior related to illegal drug injection in Mississippi was significantly more prevalent in both female students (2.6% vs. 0.9%, *p* = 0.01) and male students (4.6% vs. 1.7%, *p* < 0.001). Compared to the US, illegal drug injection prevalence in Mississippi was significantly higher for Black students (4.3% vs. 1.9%, *p* = 0.03) and Hispanic students (8.1% vs. 1.8%, *p* < 0.001), but no significant difference was observed in White students (2.0% vs. 1.1% *p* = 0.38) or in Multiple-race students (2.0% vs. 1.8%, *p* = 0.92). Compared to the US, risk behavior related to being offered, sold, or given an illegal drug on school property in Mississippi was significantly more prevalent in both female students (19.5% vs. 13.9%, *p* < 0.001) and male students (22.4% vs. 13.8%, *p* < 0.001). Compared to the US, the prevalence of being offered, sold, or given an illegal drug on school property in Mississippi was significantly higher for White students (20.6% vs. 13.2%, *p* < 0.001), Black students (19.6% vs. 13.7%, *p* = 0.01) and Hispanic students (29.2% vs. 16.7%, *p* < 0.001), but no significant difference was observed in Multiple-race students (15.8% vs. 13.8%, *p* = 0.63). 

### 3.3. Trends in Drug-Related Risk Behaviors from 2001 to 2021

Table 5 highlights the Average Annual Percent Change (AAPC) for each response to the six survey questions related to drug use, in the US and in Mississippi. Nationally, all the six drug use-related risk behaviors had a significant decrease. AAPC from 2001 to 2021 were −1.2% (*p* < 0.01), −4.2% (*p* < 0.001), −4.5% (*p* < 0.001), −8.5% (*p* < 0.001), −3.6% (*p* < 0.001) and −2.6% (*p* < 0.001), for ever having used marijuana, inhalant, heroin, methamphetamines, or injected drugs and having been offered, sold or given an illegal drug on school property, respectively. In Mississippi, however, only marijuana use showed a decrease trend, with AAPC being −1.5% (*p* <0.01). Contrary to the US trend, heroin uses, and injection of illegal drugs showed an increase pattern (*p* < 0.01 for both). Use of inhalants and methamphetamines, and ever having been offered, sold or given an illegal drug on property showed no significant change. 

Figure 1, Figure 2, Figure 3, Figure 4, Figure 5 and Figure 6 demonstrate the linear variations in the unadjusted prevalence data of drug use from 2001 to 2021 in the US vs. Mississippi. These graphs visually confirm the statistics obtained through the trend analysis, as stated earlier.

## 4. Discussion

This study highlights significant concerns regarding high-risk substance use behaviors among Mississippi teens compared to their national peers. The results of this study are consistent with other national research, suggesting that the use of most substances among teenagers has remained stable or declined [1,22,23]. Our data suggest that while national trends indicate a decrease in all six major drug-related behaviors over time, Mississippi teens reported higher instances of using inhalants, heroin, methamphetamine, and injecting illegal drugs, and being offered, sold, or given illegal drugs on school property. Factors influencing a higher rate of substance abuse among youth in Mississippi are mostly speculative, and could be poor parental monitoring, parental substance use, childhood sexual abuse, and lack of school connectedness, among others [24]. This discrepancy underscores the need for targeted interventions in Mississippi to address these alarming trends, because a strong body of research indicates that adolescent substance abuse results in negative physical and mental health consequences [3,23,24,25], as well as academic [26] and social [27] high-risk behaviors, leading to legal issues and potential criminal justice involvement [28].

### 4.1. Gender and Race

While the gender distribution of the national and Mississippi estimates was approximately equal, the racial distribution varied significantly. This difference primarily reflects the underlying racial demographics in Mississippi, where Black youth are over-represented. When comparing the types of drug abuse, inhalant use in male students in Mississippi outnumbered others, compared to the national average (12.8% vs. 6.8%, *p* <0.001). These gender- and race-specific higher rates in Mississippi of inhalant use warrant targeted intervention for these populations.

The Mississippi sample includes more 9th graders and fewer 12th graders than their national counterparts. However, the number of 10th and 11th graders was similar in both cohorts. Similar disparities in racial demographics and grade distribution have been noted in other studies. For instance, a study by Johnston et al. (2020) on substance use among adolescents highlighted the fact that demographic differences, particularly racial and educational disparities, significantly impact substance use patterns [26,28]. Additionally, the CDC’s YRBSS data suggest that regional differences, such as those seen in Mississippi, can influence the prevalence of high-risk behaviors among youth [26].

### 4.2. Drug Use-Related Comparisons

While students from both cohorts agreed with the statement about ever using marijuana in equal measure, that is where the similarities ended. In all the following categories, Mississippi youth reported significantly higher rates of substance use than their US counterparts.

In this study, we found Mississippi youth to be exhibiting dangerous, high-risk substance behaviors. When queried about inhalant use, Mississippi students were 1.5 times more likely to have tried or used inhalants than the US sample. Prevalence rates for heroin use were reported at 3.4 times the rate of the national sample. Mississippi students use methamphetamine at a rate of 2.5 times that of US youth. They reported injecting any illegal drug at 2.8 times the rate of their national peers. Finally, they were offered, sold, or given an illegal drug on school property at a rate of 1.5 times more often than the US sample. Differences in substance use behaviors between Mississippi youth and their national peers can be attributed to several factors, including socioeconomic conditions, education, and mental health resources. Socioeconomic conditions include poverty and economic hardship. Mississippi has one of the highest poverty rates in the United States, and 19.1% of its residents live below the poverty line (US 11.7%) [29]. Mississippi’s economy and healthcare system rank 50th in the US. This has resulted in its residents having the worst health outcomes, leaving Mississippians with a disproportionate disease and illness burden [30,31]. Economic hardship is strongly associated with increased substance use among adolescents. Families experiencing financial stress may have limited access to education and healthcare resources, including mental health services, which can lead to higher rates of substance abuse [32]. 

In Mississippi, educational disparities are found in preschool, where only 51% of three- and four-year-old children are enrolled in early education, and only 34% of kindergarten students score at or above state benchmarks. Three-quarters of public-school students are economically disadvantaged, with 54% of those students being African American. In 2018, 21.5% of African American students met the ACT college benchmarks in reading and math, 16 percentage points below the state average and 32 percentage points below the average score for White students [33]. Mississippi schools have fewer resources for comprehensive drug education and prevention programs when compared to schools in other states. Effective substance-abuse prevention programs are crucial in reducing the rates of adolescent drug use [34]. 

Mississippi ranks 47th in the nation for access to mental health care [35]. Mississippi is predominately a rural state [36]. Research suggests that access to mental health services is often limited in rural areas, including many parts of Mississippi. Adolescents with untreated mental health issues may turn to substance use as a form of self-medication [37]. 

### 4.3. Drug Use-Related Comparisons by Race and Gender

Upon closer inspection of the data, inhalant- and methamphetamine-use prevalence was significantly higher for Mississippi males compared to the US sample. Heroin use prevalence did not differ based on gender. However, heroin use prevalence was significantly higher for Black and Hispanic students and lower in multiple-race non-Hispanic students, with no difference for White students. However, national prevalence studies have reported higher rates of substance use in White students compared to their Black or Hispanic counterparts [32,38]. While the CDC YRBSS from 2019 also found higher substance use rates among White students, the study found that certain high-risk behaviors, such as injection drug use, were reported more frequently among some minority groups [1,39,40]. This supports our finding that both Mississippi female and male students who are Black or Hispanic reported a significantly higher prevalence of injecting illegal drugs compared to the US sample. 

Finally, high-risk behavior relating to being offered, sold, or given an illegal drug on school property in Mississippi was significantly more prevalent for both female and male White, Black, and Hispanic students when compared to the national sample. Again, research suggests structural components may contribute to these behaviors, including socioeconomic conditions, the educational disparities mentioned earlier, and regional differences in drug availability and enforcement. Economic hardship is often correlated with higher rates of substance abuse and related activities. In schools located in areas of high poverty rates, students might be exposed to drug-related activities due to the broader community context where drug dealing may be an important source of income [34,40]. Many schools in Mississippi are underfunded, lacking resources for comprehensive drug-prevention programs. This could result in decreased awareness of the risks associated with substance use [13]. The availability of drugs and regional law-enforcement practices can significantly impact the prevalence of drug-related activities in schools. Mississippi might have counties where drug trafficking is more prevalent, increasing the likelihood of students being exposed to drugs. Furthermore, differences in local law-enforcement practices can affect how drug-related activities are monitored and controlled within school environments. 

### 4.4. Limitations

YRBSS data focus on teens attending school. However, they do not account for students who have dropped out of high school, were homeschooled, attended alternative schools, or were part of the estimated 5% of youth who have never attended school [41]. According to the US Department of Education, in 2001, the high-school dropout rate was 9.9% [41]. Of Mississippi high school students entering school in 2001, the dropout rate by 2004 was 26%. The Mississippi dropout rates did not fall to 10% until 2019 [42]. This cross-sectional study also makes determining whether students are over- or under-reporting health-related behaviors impossible. 

We did not include prescription opiates in our study. This information would have enhanced our understanding of recent trends in prescription drug use and the opioid epidemic. The YRBSS does not include electronic vape pen use, which has become increasingly popular with teens [43,44]. Our data suggest that polysubstance use needs an additional focus, as new studies are providing compelling evidence that abusing opioid prescription drugs increases the likelihood of dangerous polydrug abuse [45]. 

## 5. Conclusions

An important finding in this study is that while overall teen substance use is decreasing nationwide, in Mississippi this is not the case. While Mississippi high school-student marijuana use is declining slightly, they are significantly more likely to use inhalants, methamphetamine, and heroin, and to practice high-risk behaviors, including injecting drugs and accessing drugs at school, than their national peers. We must find ways in which to intervene. Mississippians face far greater rates of poverty and lack of access to health and mental health services due to inequities in the healthcare system. As a result, Mississippi high school students who use drugs are at risk of a bevy of negative outcomes, including unprotected sex, sexually transmitted infections, and teen pregnancy, in a state where reproductive health is severely limited, and infant and mortality deaths occur at twice the national average. 

Because of the significant prevalence of illicit drug use among Mississippi adolescents, there is an urgent need for an integrated approach to reverse the rapidly growing problem. Public health actions, such as school-based prevention programs, community outreach initiatives, peer and family counseling, drug-free zones or policy reforms could be impactful for curving substance abuse in adolescents. 

## Figures and Tables

**Figure 1 ijerph-21-00919-f001:**
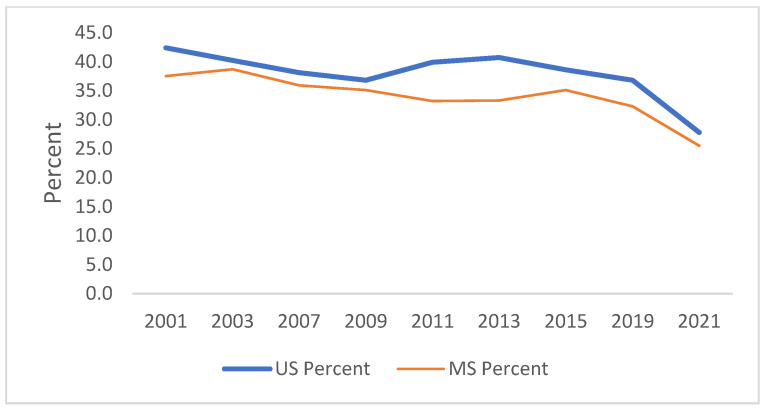
Trend in ever used marijuana in lifetime.

**Figure 2 ijerph-21-00919-f002:**
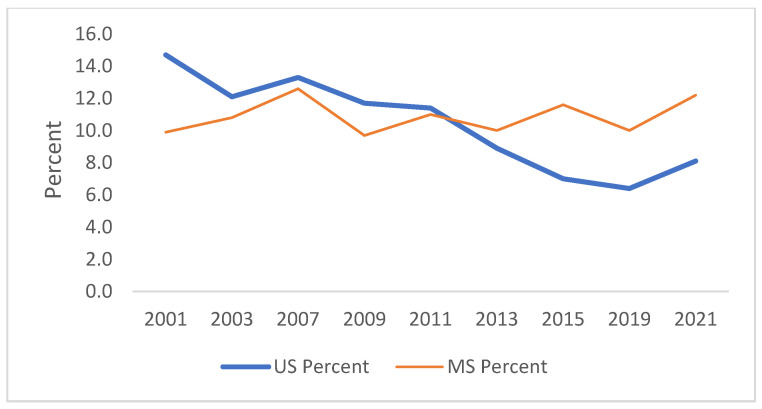
Trend in ever used inhalants in lifetime.

**Figure 3 ijerph-21-00919-f003:**
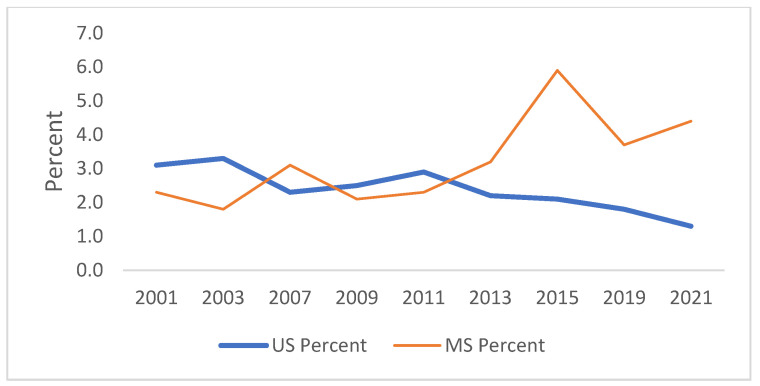
Trend in ever used heroin in lifetime.

**Figure 4 ijerph-21-00919-f004:**
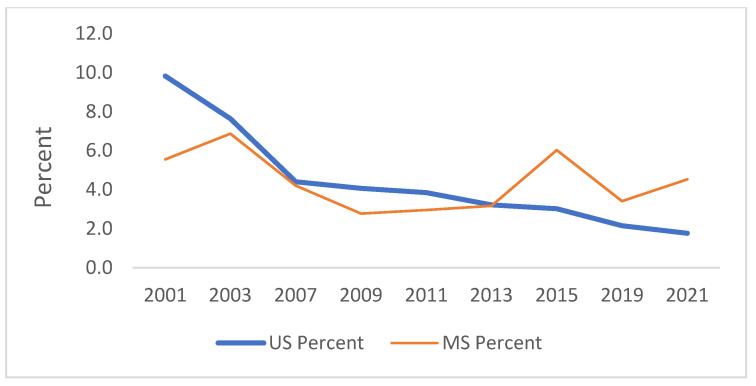
Trend in ever used methamphetamines in lifetime.

**Figure 5 ijerph-21-00919-f005:**
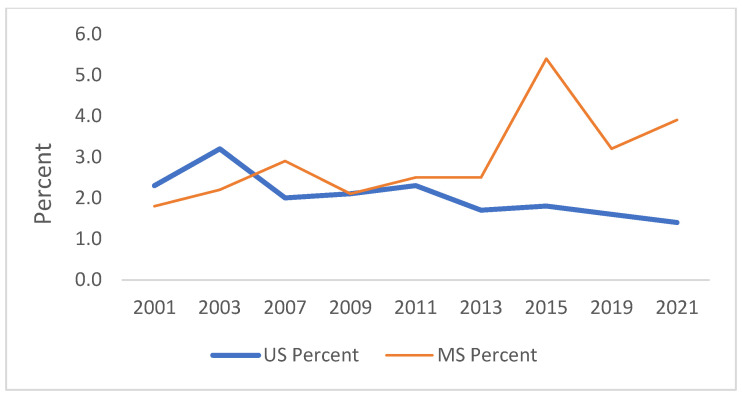
Trend in ever injected illegal drugs in lifetime.

**Figure 6 ijerph-21-00919-f006:**
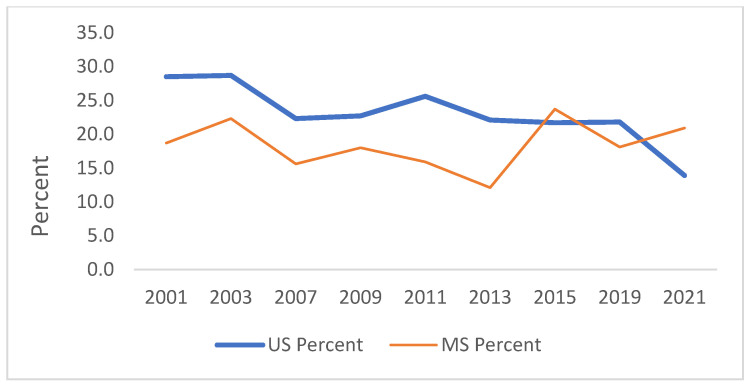
Trend in offered, sold or given an illegal drug on school property.

**Table 1 ijerph-21-00919-t001:** Comparison of prevalence of drug use-related risk behaviors between Mississippi and the United States, YRBSS 2021.

Risk Behavior	Mississippi (%)	US (%)	Prevalence Ratio (95% *CI*) *	*p*-Value
Ever used marijuana	25.5	27.8	0.9 (0.8, 1.0)	0.19
Ever used inhalants	12.2	8.1	1.5 (1.2, 1.8)	<0.001
Ever used heroin	4.4	1.3	3.3 (2.3, 4.9)	<0.001
Ever used methamphetamines	4.5	1.8	2.6 (1.8, 3.7)	<0.001
Ever injected any illegal drug	3.9	1.4	2.7 (1.9, 3.8)	<0.001
Were offered, sold or given an illegal drug on school property	20.9	13.9	1.5 (1.3, 1.7)	<0.001

* Prevalence ratio is MS prevalence over the US prevalence.

**Table 2 ijerph-21-00919-t002:** Prevalence of marijuana and inhalant use in Mississippi and the United States, YRBSS 2021.

Variable	Ever Used Marijuana, Prevalence (95% *CI*), *n*			Ever Used Inhalant, Prevalence (95% *CI*), *n*		
	Mississippi	US	*p*-Value	Mississippi	US	*p*-Value
Overall	25.5 (22.8, 28.3) 1622	27.8 (25.5, 30.3) 13,899	0.19	12.2 (10.1, 14.5) 1682	8.1 (7.4, 8.9) 9911	<0.001
Gender						
Female	27.2 (23.8, 30.8) 819	30.9 (28.0, 33.8) 6544	0.10	10.6 (7.8, 14.4) 835	9.4 (8.4, 10.5) 4686	0.47
Male	23.7 (20.0, 27.8) 787	24.8 (22.6, 27.1) 7146	0.61	12.8 (10.4, 15.7) 825	6.8 (6.0, 7.8) 5075	<0.001
Race						
White	23.4 (18.5, 29.1) 576	26.2 (24.3, 28.1) 7754	0.31	11.1 (8.2, 14.8) 584	8.3 (7.1, 9.6) 4897	0.11
Black	26.5 (22.9, 30.4) 739	33.3 (28.2, 38.9) 1696	0.06	11.8 (8.9, 15.4) 774	7.0 (6.0, 8.3) 1496	<0.05
Hispanic/Latino	25.4 (17.8, 34.8) 143	31.2 (27.6, 35.1) 2552	0.20	15.7 (9.3, 25.4) 152	8.2 (7.1, 9.5) 2112	0.06
Multiple race	27.1 (16.4, 41.4) 55	35.2 (27.5, 43.6) 745	0.27	8.1 (3.2, 19.1) 57	10.8 (7.6, 15.1) 563	0.49

**Table 3 ijerph-21-00919-t003:** Prevalence of heroin and methamphetamines use in Mississippi and the United States, YRBSS 2021.

Variable	Ever Used Heroin, Prevalence (95% *CI*), *n*			Ever Used Methamphetamines, Prevalence (95% *CI*), *n*		
	Mississippi	US	*p*-Value	Mississippi	US	*p*-Value
Overall	4.4 (3.1, 6.3) 1671	1.3 (1.1, 1.6) 16,837	<0.001	4.5 (3.2, 6.3) 1695	1.8 (1.5, 2.1) 16,817	<0.001
Gender						
Female	2.3 (1.3, 3.9) 830	0.8 (0.5, 1.2) 7977	0.02	2.7 (1.5, 4.6) 835	1.4 (1.1, 1.8) 7972	0.08
Male	5.9 (4.0, 8.7) 818	1.6 (1.3, 2.1) 8604	<0.001	5.6 (4.1, 7.7) 837	1.9 (1.4, 2.4) 8588	<0.001
Race						
White	1.5 (0.4, 5.5) 581	1.0 (0.7, 1.3) 9012	0.61	2.3 (1.0, 5.2) 585	1.4 (1.1, 1.8) 9007	0.33
Black	5.8 (3.8, 8.7) 767	1.7 (1.1, 2.6) 2213	<0.001	5.4 (3.5, 8.3) 781	2.0 (1.4, 2.9) 2205	<0.001
Hispanic/Latino	8.9 (4.1, 18.2) 149	1.6 (1.2, 2.1) 3156	0.03	10.7 (5.7, 19.3) 155	2.3 (1.8, 2.9) 3150	0.01
Multiple race	0.0 (0.0, 0.0) 57	1.5 (0.8, 2.8) 972	<0.001	2.0 (0.3, 13.5) 57	2.2 (1.3, 3.7) 970	0.92

**Table 4 ijerph-21-00919-t004:** Prevalence of illegal drug injection and drug on school property in Mississippi and the United States, YRBSS 2021.

Variable	Ever Injected any Illegal Drug, Prevalence (95% *CI*), *n*			Ever Offered, Sold, or Given an Illegal Drug on School Property, Prevalence (95% *CI*), *n*		
	Mississippi	US	*p*-Value	Mississippi	US	*p*-Value
Overall	3.9 (2.8, 4.9) 1674	1.4 (1.1, 1.8) 12,965	<0.001	20.9 (18.9, 23.1) 1665	13.9 (13.1, 14.8) 16,366	<0.001
Gender						
Female	2.6 (1.7, 4.0) 829	0.9 (0.6, 1.5) 6196	0.01	19.5 (17.2, 22.1) 829	13.9 (12.5, 15.4) 7797	<0.001
Male	4.6 (3.2, 6.5) 824	1.7 (1.3, 2.2) 6576	<0.001	22.4 (18.4, 26.9) 816	13.8 (12.9, 14.8) 8320	<0.001
Race						
White	2.0 (0.7, 5.6) 581	1.1 (0.7, 1.7) 6224	0.38	20.6 (16.6, 25.2) 583	13.2 (12.0, 14.6) 8868	<0.001
Black	4.3 (2.6, 5.9) 774	1.9 (1.1, 2.8) 2007	0.03	19.6 (16.8, 22.3) 770	13.7 (11.0, 16.5) 2214	0.01
Hispanic/Latino	8.1 (4.6, 13.8) 151	1.8 (1.3, 2.5) 2749	<0.001	29.2 (24.0, 35.1) 147	16.7 (15.3, 18.2) 3011	<0.001
Multiple race	2.0 (0.3, 13.5) 55	1.8 (1.0, 3.5) 787	0.92	15.8 (9.8, 24.3) 56	13.8 (9.9, 18.8) 918	0.63

**Table 5 ijerph-21-00919-t005:** Trend in drug-use risk behaviors among Mississippi and the US adolescents, YRBSS 2001–2021.

Variable	Location	Adjusted Prevalence (%) *		*p*-value				AAPC % (95% *CI*)	Trend
		2001	2021	Linear **	Quadratic **	Cubic **	Jointpoint ***		
Ever used marijuana	United States	43.3	27.3	0.00	0.00	0.00	<0.01	−1.2 (−2.0, −0.4)	Significant decrease
	Mississippi	37.5	24.8	0.00	0.00	0.00	<0.01	−1.5 (−2.4, −0.7)	Significant decrease
Ever used inhalant	United States	14.6	8.0	0.00	0.27	0.00	0.00	−4.2 (−5.9, −2.4)	Significant decrease
	Mississippi	9.9	11.0	0.56	0.75	0.09	N/A	N/A	No significant change
Ever used heroin	United States	3.3	1.1	0.00	0.03	0.04	0.00	−4.5 (−6.1, −2.9)	Significant decrease
	Mississippi	2.4	3.5	<0.01	0.77	0.06	N/A	3.6 (−0.5, 7.9)	Significant increase
Ever used methamphetamines	United States	10.1	1.5	0.00	0.11	0.06	0.00	−8.5 (−9.3, −7.7)	Significant decrease
	Mississippi	5.6	3.7	0.01	0.03	0.60	0.18, 0.78	N/A	No significant change
Ever injected any illegal drug	United States	2.4	1.2	0.00	0.79	0.74	0.00	−3.6 (−4.9, −2.3)	Significant decrease
	Mississippi	1.8	3.2	<0.01	0.51	0.39	N/A	3.2 (−0.6, 7.2)	Significant increase
Were offered, sold, or given an illegal drug on school property	United States	29.2	13.5	0.00	0.03	0.00	<0.01	−2.6 (−3.9, −1.3)	Significant decrease
	Mississippi	18.7	20.0	0.9	0.00	0.8	0.50, 0.57	N/A	No significant change

* Adjusted for sex, race and grade; AAPC—average annual percent change calculated using joinpoint regression. ** Results of logistic regression models; *** two or more joinpoint *p*-values are for the segments resulting from significant quadratic or cubic changes; N/A means not available.

## Data Availability

Publicly archived datasets from Centers for Disease Control and Prevention were analyzed or generated during the study. Data are available at the following link: https://www.cdc.gov/healthyyouth/data/yrbs/index.htm, (accessed on 10 June 2024).

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
