# Peer review of "The Prevalence of and Trend in Drug Use among Adolescents in Mississippi and the United States: Youth Risk Behavior Surveillance System (YRBSS) 2001–2021"

_ijerph, 2024, doi:10.3390/ijerph21070919_

Round 1

Reviewer 1 Report

Comments and Suggestions for Authors

Thank you for giving me the opportunity to review this interesting manuscript. In my opinion, the manuscript appears readable, coherent, and thoroughly addresses the issues related to substance use. However, some revisions are necessary:

Introduction

The introduction provides a clear rationale for the study, emphasizing the demographic uniqueness of Mississippi compared to the national average.

 However, it could benefit from more detailed background information on previous findings related to adolescent drug use trends. In the introduction, I would also include the risk of substance-induced psychosis, which has recently become an important clinical emergency. (cit https://doi.org/10.1016/j.psychres.2023.115053; https://doi.org/10.1016/j.biopsych.2015.08.001)

Objective:

 The objectives are clearly stated. They outline the purpose of examining drug use trends and differences by gender and race.

Methods

The study design is appropriate for the research question. It is clear that data from the YRBSS from 2001 to 2021 were used. The statistical methods are well-described, including summary statistics, prevalence ratios, and chi-squared tests.

However, a brief explanation of why these particular methods were chosen would enhance the clarity.

The methods section would benefit from additional detail to ensure reproducibility. For example, specify how the data were cleaned and any assumptions made during analysis.

Results

Results are presented clearly and the interpretation is logical  but authors should more emphasis on the significance of the findings in the broader context of public health.

Moreover while the results are comprehensive, ensure that all key findings are explicitly stated. The current presentation sometimes requires the reader to infer significance from the data.

Discussion

The discussion section effectively contextualizes the findings within the existing literature. However, it could further explore potential explanations for the observed trends and differences.

The public health implications are appropriately discussed in the manuscript. The study effectively highlights the importance of understanding drug use trends among adolescents in Mississippi and the United States. I suggest that the authors strengthen the section by offering more specific recommendations based on the study's findings. For example, if certain drugs are found to be more prevalent among specific demographic groups, it would be beneficial to suggest targeted intervention programs for those groups. If the study identifies particular years or periods with significant changes in drug use trends, discussing potential public health campaigns or policy changes that could have influenced these trends would be valuable. Providing concrete suggestions for public health actions, such as school-based prevention programs, community outreach initiatives, or policy reforms, would make the implications of your findings more actionable and impactful.

Conclusion

The conclusion effectively summarizes the main findings and their implications, providing a clear and concise wrap-up of the study. I suggest avoid introducing any new information that has not been covered in the results or discussion sections. Introducing new data or insights in the conclusion can confuse readers and detract from the coherence of the manuscript. Ensure that all major findings and interpretations are fully discussed in the earlier sections of the paper.

The manuscript may already hint at some implications or future directions, but explicitly suggesting specific areas for future research would greatly enhance the conclusion, for  example, if certain trends were identified but not fully explained, recommend targeted studies to explore these trends in more depth. If your study reveals gaps in the existing data or knowledge, point these out and suggest how future research could address them.

Summary

This manuscript presents valuable insights into adolescent drug use trends over two decades, comparing Mississippi with national data. While the study is well-conceived and the manuscript is well-structured, several areas could be improved for clarity, depth, and impact. Addressing the noted weaknesses will enhance the overall quality and readability of the manuscript, making it suitable for publication and aligning it with the journal's standards.

Reviewer 2 Report

Comments and Suggestions for Authors

The aim of the study was to examine the drug use among adolescents in Missisipi compared to that in the US, which included determining prevalence and trends in drug use as well as drug on school property. 

This research showed detailed findings on drug use-related issues in Missisipi, which can be considered quite alarming. This poses an important challenge for public health in Missisipi. 

Please provide formulas of the logistic and jointpoint regression models in supplementary material.

Figures 2-6 - please use "US percent" and "MS percent" in legends for greater readibility.

line 17 - abbreviation YRBSS not explained

line 69 - abbreviation YRBSS not explained

line 83 - missing period at the end of the sentence

line 96 - "there 95% confidence intervals", the correct word is "their" 

line 176 Table 3 and line 172 - different valeus in text and in table for Black students

line 231 - abbreviation YRBSS already introduced

line 295 - white should be capitalized

line 309 - reference missing

Round 2

Reviewer 1 Report

Comments and Suggestions for Authors

The authors have responded appropriately to my requests.